# FOXM1: A Multifunctional Oncoprotein and Emerging Therapeutic Target in Ovarian Cancer

**DOI:** 10.3390/cancers13123065

**Published:** 2021-06-19

**Authors:** Cassie Liu, Carter J. Barger, Adam R. Karpf

**Affiliations:** Eppley Institute and Fred & Pamela Buffett Cancer Center, University of Nebraska Medical Center, Omaha, NE 68918-6805, USA; cassie.liu@unmc.edu (C.L.); Carter.barger@gmail.com (C.J.B.)

**Keywords:** FOXM1, forkhead box M1, transcription factors, oncogenes, oncoproteins, ovarian cancer, high-grade serous ovarian cancer

## Abstract

**Simple Summary:**

Ovarian cancer is a lethal disease in women with a 10-year survival rate of <40% worldwide. A key molecular alteration in ovarian cancer is the aberrant overexpression and activation of the transcription factor forkhead box M1 (FOXM1). FOXM1 regulates the expression of a multitude of genes that promote cancer, including those that increase the growth, survival, and metastatic spread of cancer cells. Importantly, FOXM1 overexpression is a robust biomarker for poor prognosis in pan-cancer and ovarian cancer. In this review, we first discuss the molecular mechanisms controlling FOXM1 expression and activity, with a specific emphasis on ovarian cancer. We then discuss the evidence for and the manner by which FOXM1 expression promotes aggressive cancer biology. Finally, we discuss the clinical utility of FOXM1, including its potential as a cancer biomarker and as a therapeutic target in ovarian cancer.

**Abstract:**

Forkhead box M1 (FOXM1) is a member of the conserved forkhead box (FOX) transcription factor family. Over the last two decades, FOXM1 has emerged as a multifunctional oncoprotein and a robust biomarker of poor prognosis in many human malignancies. In this review article, we address the current knowledge regarding the mechanisms of regulation and oncogenic functions of FOXM1, particularly in the context of ovarian cancer. FOXM1 and its associated oncogenic transcriptional signature are enriched in >85% of ovarian cancer cases and FOXM1 expression and activity can be enhanced by a plethora of genomic, transcriptional, post-transcriptional, and post-translational mechanisms. As a master transcriptional regulator, FOXM1 promotes critical oncogenic phenotypes in ovarian cancer, including: (1) cell proliferation, (2) invasion and metastasis, (3) chemotherapy resistance, (4) cancer stem cell (CSC) properties, (5) genomic instability, and (6) altered cellular metabolism. We additionally discuss the evidence for FOXM1 as a cancer biomarker, describe the rationale for FOXM1 as a cancer therapeutic target, and provide an overview of therapeutic strategies used to target FOXM1 for cancer treatment.

## 1. Introduction

### 1.1. FOX Proteins and FOXM1 Discovery

Forkhead box (FOX) proteins are a family of transcription factors defined by an evolutionarily conserved 80–100 amino acid domain called the forkhead box or winged-helix motif [1,2,3,4,5]. There are 50 known FOX genes in the human genome divided into 19 subfamilies (A–S) [6]. Sequence variation outside of the shared forkhead box motif leads to diverse mechanisms of regulation and function between FOX subfamilies, and different FOX subfamilies are involved in critical cellular processes and govern organ system functions [7]. For example, the FOXA subfamily participates in cellular metabolism and the development of endoderm-derived organs [8]; the FOXC subfamily promotes blood vessel maturation and lymphatic sprouting [9]; the FOXO subfamily regulates cell cycle arrest at G1, apoptosis, and resistance to oxidative and cellular stress [10]; the FOXP subfamily is involved in immune system functions, including coordinating the development and function of B and T lymphocytes [11].

FOXM1 is the sole member of the FOXM subfamily [6]. Historically, FOXM1 held alternative names: Trident [12], WIN [13], hepatocyte nuclear factor 3 (HNF3) [14], fork head homolog 11 (HFH-11) [14,15], FKHL16 [15], M-phase phosphoprotein 2 (MPHOSPH2, MPP2) [16], MPM2-reactive phosphoprotein 2 (MPP2) [16], and TGT3 [17]. In 1997, FOXM1 was first identified by Korver et al. as Trident from mouse thymus tissue [12], by Yao et al. as WIN from a rat insulinoma cell line [13], and by Ye et al. as HNF-3/HFH-11 from a human colon carcinoma cell line [14]. These initial studies reported that FOXM1 is widely expressed in embryonic tissues but that its expression is restricted in adult tissues to actively proliferating cells, such as those in the thymus and gastrointestinal tract [12,13,14,18]. Indeed, later studies confirmed that FOXM1 is highly expressed in embryonic [19,20,21,22], regenerative [23,24,25,26], and cancerous tissues [27,28,29] and all of them exhibit high proliferative capacity. The FOXM1 field and publications have increased steadily since the year 2000, with a large proportion of studies focused on cancer (Figure 1).

### 1.2. FOXM1 Structure and Transcriptional Activity

The human *FOXM1* gene, located at chromosome 12p13.33, consists of ten exons, including exons I–VIII and alternatively spliced exons Va and VIIa [13,15,28]. Exon Va encodes a 15 amino acid insertion into the DNA binding domain (DBD), which is a feature not seen in other FOX family members. On the other hand, exon VIIa encodes a 38 amino acid insertion into the transactivation domain (TAD) [13,14,28]. Alternative splicing of exons Va and VIIa gives rise to three well-characterized FOXM1 isoforms (Table 1): FOXM1a includes all ten exons, FOXM1b omits alternative exons Va and VIIa, and FOXM1c includes alternative exon Va and omits alternative exon VIIa [13,14,28,29]. Notably, functional characterizations have shown that FOXM1b and FOXM1c are transcriptionally active while FOXM1a is not [12,14,27,28,31,32]. Additional FOXM1 isoforms have also been reported in the literature but have not been well characterized to date (Table 1).

FOXM1 contains three recognized functional protein domains: (1) a negative regulatory domain (NRD) at the N-terminus, (2) a centrally located DBD, and (3) an acidic TAD at the C-terminus (Figure 2) [12,15,37,38]. As a transcription factor, FOXM1 binds DNA within the major groove at promoters containing the canonical forkhead motif RYAAAYA, where R = purine (A/G) and Y = pyrimidine (C/T) [12,39]. FOXM1 appears to bind to its recognition sequence with lower affinity than other FOX family members; this suggests that FOXM1 may utilize more complex mechanisms for DNA binding and gene expression regulation [39]. For example, FOXM1 can be recruited to gene targets by either directly binding Forkhead consensus sequences [40] or indirectly through protein–protein interactions with the MuvB complex (LIN9, LIN37, LIN52, LIN54, and RBBP4) [41,42]. In addition, FOXM1 interaction with MuvB and B-Myb can facilitate FOXM1 binding directly to non-consensus genomic sequences [41,42,43]. Recently, an investigation using FOXM1 chromatin immunoprecipitation sequencing (ChIP-Seq) in five cancer cell lines from different organ origins revealed that FOXM1 can interact directly or indirectly with the Nuclear Transcription Factor Y (NFY) complex to regulate the majority of its cell cycle- and mitosis-related gene targets [44]. This study also proposed that FOXM1 may promote cell type-specific gene expression through additional mechanisms, such as binding to super-enhancers and interacting with cell type-specific transcription factors [44]. In agreement, FOXM1 was recently identified as a key transcriptional regulator of cancer-specific enhancers in lung adenocarcinoma [45].

### 1.3. FOXM1 Function and Regulation

FOXM1 exhibits spatiotemporal expression and activity throughout the cell cycles. FOXM1 mRNA and protein expression increases in late G1-phase, peaks in S-phase, and remains at high levels in G2/M through late M-phase [12,28,46]. In addition to regulation at the transcriptional and translational levels, fine-tuning of FOXM1 expression also occurs post-translationally. FOXM1 protein stability is regulated by the E3 ubiquitin ligase complex CRL4^VprBP^ in G_1_-phases and S-phases [47] and by the F-box protein FBXO31 during the G_2_-M transition [48]. Another critical regulator of FOXM1 function is post-translational modification by phosphorylation. During cell cycle progression, FOXM1 undergoes sequential multi-site phosphorylation by several cyclin-CDK complexes (Cyclin D-CDK4/6 [49], Cyclin E-CDK2 [50], and Cyclin A/B-CDK1/2 [51,52,53,54]) as well as by other kinases (CHK2 [55], MAPK [56], and PLK1 [57]), all of which facilitate FOXM1 stabilization, nuclear translocation, and activation [29]. Upon mitotic exit, FOXM1 is targeted for ubiquitin-mediated proteasomal degradation by the anaphase promoting complex/cyclosome (APC/C) E3 ubiquitin ligase [58]. Thus, the cycle of FOXM1 expression, phosphorylation, and degradation repeats with every cell division cycle.

The NRD of FOXM1 binds to the TAD to repress FOXM1 transcriptional activity (auto-repression). Importantly, phosphorylation of the TAD relieves self-inhibition from the NRD, which allows for differential activation and gene targeting of FOXM1 at different cell cycle stages [37,50,52,54,59]. A recent investigation supports a model where FOXM1 phosphorylation by CDK proteins provides a docking site for polo-like kinase 1 (PLK1), which then phosphorylates the TAD at S730 on FOXM1c (S715 on FOXM1b) and releases it from the NRD [59]. The TAD and NRD become structurally disordered upon disassociation and offer flexibility in interacting with binding partners such as p300/CBP [59]. This model may be primarily applicable to FOXM1 activation at the G2/M checkpoint, where PLK1 is known to function. Thus far, no other kinase has been shown to phosphorylate the TAD at S730 on FOXM1c (S715 on FOXM1b).

Once activated, FOXM1 promotes entry into S phase by activating the transcription of genes regulating the G1/S checkpoint (e.g., *SKP2* and *CKS1*) [60]. Later, entry into M phase is mediated by FOXM1 activation of genes regulating the G2/M checkpoint (e.g., *PLK1*, *CDC25B*, *CCNB1*, *NEK2*, and *BIRC5*) [60,61,62]. Finally, FOXM1 promotes faithful mitotic progression by activating genes involved in mitotic spindle assembly and chromosome segregation (e.g., *AURKB*, *KIF20A*, *CENPA*, *CENPB*, and *CENPF*) [60,61,62]. Thus, FOXM1 functions as a critical transcriptional regulator of several important cell cycle transitions, and its ability to differentially activate gene sets may relate both to its sequential phosphorylation as well as the presence of distinct binding partner complexes at different cell cycle phases.

In addition to phosphorylation and ubiquitination, FOXM1 is regulated by other post-translational modifications [29]. Small ubiquitin modifiers (SUMOs) may be necessary for both FOXM1 activation and degradation, depending on the SUMOylation site [63,64,65,66,67]. Acetylation of FOXM1 may promote its activity [68], while methylation of FOXM1 may suppress its activity [69]. SUMOylation, acetylation, and methylation of FOXM1 require further study to discern their relative roles and potential crosstalk in FOXM1 regulation.

### 1.4. Transgenic Mouse Models Reveal Functions of FOXM1

Whole-body knockout of FOXM1 results in embryonic lethality in mice. *Foxm1*^−/−^ mouse embryos with deletion of exons IV–VII, which removes the DBD and TAD, died in utero between E13.5 and E16.5 due to a failure to properly form multiple organs, including the liver, lung, and heart [20,21,22]. Notably, polyploidy was observed in hepatoblasts and embryonic cardiomyocytes, which likely results from improper mitotic progression [20,22]. *Foxm1*^−/−neo^ mice, where the PGK-neomycin cassette was inserted into Exon III of *Foxm1*, died perinatally with increased numbers of polyploid cells in the developing heart and liver, which is a phenomenon observed as early as E13 [70]. Thus, whole-body knockout revealed the importance of FOXM1 expression in early organogenesis due to the lethal phenotypes observed. These models did not allow investigation of post-embryonic functions of FOXM1.

Conditional *Foxm1* knockout (cKO) mice have been used to investigate the role of FOXM1 in tissue-specific organogenesis [71]. For example, *Foxm1* deletion in smooth muscle cells did not influence cell differentiation but, rather, decreased proliferation of smooth muscle cells in blood vessels and the esophagus [72]. Deletion of *Foxm1* in respiratory epithelial cells prenatally impaired important lung functions (sacculation, type I cell differentiation, and surfactant production) but did not alter gross lung morphology [73]. Conversely, postnatal expression of constitutively active FOXM1b in respiratory epithelial cells increased Clara cell proliferation and airway hyperplasia [74]. Deletion of *Foxm1* in pancreatic tissue resulted in a normal pancreas at birth but with a lack of β-cell mass expansion postnatally, which can result in impaired islet function and overt diabetes [75].

Transgenic mouse models have also been utilized to investigate the roles of FOXM1 in organ regeneration in adult tissues [71]. In response to vascular injury, mice with *Foxm1* cKO in endothelial cells showed difficulty reannealing adherens junctions, microvessel leakage, and poor endothelial barrier function [76,77]. In the setting of liver regeneration, hepatocyte-specific *Foxm1* deletion slowed hepatocyte proliferation [78], while hepatocyte-specific FOXM1b overexpression led to accelerated hepatocyte growth through increased S-phase and M-phase transitions [46,79,80]. In mice challenged by lung injury, FOXM1b overexpression increased the proliferation of several cell types in the lung [81], while pancreas-wide *Foxm1* deletion [82] and *Foxm1* deletion in muscle satellite cells [26] led to impairments in pancreas and muscle repair, respectively, following injury. Collectively, studies using transgenic mice with *Foxm1* KO, *Foxm1* cKO, and *FOXM1b* overexpression verify an important role for FOXM1 in embryogenesis, organogenesis, and organ regeneration in adults.

Transgenic mouse models have also been utilized to define the functions of FOXM1 in cancer [71]. FOXM1 overexpression in all cell types through a *Rosa26-FOXM1b* construct, when used in combination with a second tumor induction stimulus or oncogene, increased the number and size of lung adenomas [83]; increased the number, size, and proliferation of colon adenocarcinomas [84]; and accelerated the development of prostate adenocarcinomas [85]. In a transgenic mouse model expressing a constitutively active form of FOXM1b (*FOXM1-∆N*), FOXM1 stimulated progression from urethane-induced benign lung adenomas to invasive metastatic lung adenocarcinomas [86]. However, in instances where FOXM1 was tested as the main tumor inducer (i.e., without a second tumor induction stimulus or oncogene), *FOXM1b* overexpression alone was insufficient to generate hepatocellular carcinoma [87] and lung adenocarcinoma [74]. Deletion of *Foxm1* prior to or following a tumor induction stimulus in mice decreased both the number and size of tumors in the lung [88,89], liver [90], and colon [84]. Interestingly, although FOXM1 overexpression alone appears to be insufficient in driving tumorigenesis in vivo, the absence of FOXM1 prevented tumor development in the colon [84] and liver [91]. These data suggest that FOXM1 is necessary but not sufficient for tumorigenesis in vivo.

To date, transgenic mouse models have not been utilized to interrogate the oncogenic function of FOXM1 in ovarian cancer. This is partially due to the relative dearth of transgenic ovarian cancer models. However, given the recent development of such models [92,93,94], this is an area that is ripe for future investigations.

## 2. Ovarian Cancer

Ovarian cancer is the fifth leading cause of cancer-related deaths in women in the United States [95] and the eighth leading cause of cancer-related deaths in women globally [96]. Due to the internal location of the ovaries and non-specific symptoms during early-stage disease, ovarian cancer is difficult to detect early on and is frequently diagnosed during the advanced clinical stages (III and IV) [97]. The standard of care treatment for ovarian cancer is cytoreductive (debulking) surgery, platinum-based and taxane combination chemotherapy, and maintenance therapy using poly (ADP-ribose) polymerase inhibitors (PARPi) and/or bevacizumab (an inhibitor of vascular endothelial growth factor (VEGF)) [98,99]. However, >80% advanced-stage ovarian cancer eventually recurs and treatment with further therapy is palliative [100]; this results in a 10-year overall survival of <40% worldwide [101,102]. The lethality of ovarian cancer highlights the need to develop better early detection modalities and more effective therapies for relapsed patients.

There are three major categories of ovarian cancer: epithelial (EOC), germ cell, and sex cord-stromal [100]. Of these, EOC comprises ~90% of all ovarian cancer cases [100] and is the focus of this review article. EOC is subtyped by histology into serous (low-grade or high-grade), endometroid, mucinous, and clear cell tumors [100]. Ovarian tumors with serous histology constitute ~70% of EOC, with most cases being high-grade serous carcinoma (HGSC) [103]. Furthermore, HGSC accounts for up to 90% of ovarian cancer-related deaths overall [104]. Although HGSC was traditionally thought to arise from the ovarian surface epithelium (OSE), a majority of the evidence now supports the contention that the fallopian tube epithelium (FTE) is the origin of most HGSC [105,106,107]. Notably, crucial studies identified HGSC precursors, mainly serous tubal intraepithelial carcinomas (STICs), in the fallopian tubes (FT) of women with disseminated HGSC [108,109,110,111] and in those at high risk for developing HGSC (e.g., *BRCA1/2* mutation carriers) [112,113,114].

The prevailing model for HGSC development proposes that normal FTE transforms into early serous proliferations (ESPs) and then STICs, which ultimately become HGSC that disseminate to the ovaries and/or other sites in the peritoneal cavity [105]. Accordingly, ESPs, also known as “p53 signature” lesions, show increased expression of mutant p53 and γ-H2AX (an indicator of DNA damage) compared to normal FTE [112,115]. STICs show an atypical morphology in addition to the high expression of mutant p53, γ-H2AX, and Ki-67 (an indicator of proliferative activity) [112,115]. HGSC maintains high expression of mutant p53, γ-H2AX, and Ki-67 and experiences profound genomic instability with recurrently altered pathways, including p53, Rb/E2F, homologous recombination (HR) DNA repair, PI3K/RAS, NOTCH, and FOXM1 [30,115,116].

## 3. FOXM1 Is Overexpressed and Activated in Ovarian Cancer

In 2011, The Cancer Genome Atlas (TCGA) published an integrated genomic analysis of primary HGSC tumors. This landmark study identified FOXM1 as a key oncoprotein in ovarian cancer [30]. The FOXM1 transcriptional pathway was aberrantly activated in over 85% of cases and rendered it the second most frequent molecular alteration in HGSC, second only to *TP53* mutations [30]. Publications on FOXM1 in ovarian cancer have subsequently increased since 2011 (Figure 1). Supporting the initial conclusions of the TCGA, other studies have described widespread overexpression of FOXM1 and its transcriptional targets in HGSC and EOC and have explored the molecular mechanisms underlying FOXM1 pathway activation. These mechanisms are described in detail below and are summarized in Figure 3.

### 3.1. The FOXM1 Gene Experiences Copy Number Gains and Amplifications

Chromosome 12p13.33, where FOXM1 resides, shows copy number gains and amplifications in many human cancers, and studies indicate that this somatic copy number alteration (SCNA) can occur early in cancer development [117,118,119,120,121,122,123,124,125,126]. We reported that, in TCGA data, 45% of HGSC harbored low-level copy number gain of FOXM1, while 12% of tumors harbored high-level amplification, impacting almost 60% of cases [127]. Importantly, FOXM1 copy number correlated with mRNA expression in HGSC and this demonstrates that FOXM1 copy number gain is a functional SCNA [127]. In agreement, we more recently showed that *FOXM1* copy number correlates with FOXM1 mRNA and protein expression in pan-cancer [27]. The 12p13.33 amplifications were reported to be common in an independent cohort of HGSC patients before and after front-line chemotherapy (and detectable in plasma), further supporting this mechanism [128]. Despite the high frequency of FOXM1 SCNA observed in HGSC, an even greater proportion of tumors show FOXM1 overexpression or pathway activation (reaching ~90% of HGSC cases) [30,127], indicating that additional mechanisms contribute to FOXM1 overexpression and pathway activation in HGSC.

### 3.2. Inactivation of Upstream Tumor Suppressor Promotes FOXM1 Gene Expression

The *TP53* mutations are ubiquitous in HGSC [30,129]. Mutations in *TP53* frequently manifest as single base-pair mutations localized to the DBD, resulting in either partial or complete loss of function (LOF) or gain-of-function (GOF) [130]. GOF p53 mutants exhibit both loss of wild-type and gain of oncogenic functions, leading to the promotion of cell proliferation, survival, migration, invasion, and angiogenesis [130,131]. Important early studies revealed wild-type p53 as a negative regulator of FOXM1 [132,133]. In HGSC, bioinformatic pathway prediction suggested that dysfunctional p53 is linked to the upregulation of *FOXM1*, *VEGFA*, *TPX2*, *BRIC5*, and *TOP2A* [134]. Moreover, GOF p53^R273H^ and p53^R248W/Q^, which are three of the most common p53 mutations in ovarian cancer [130,131], were noted to dramatically increase FOXM1 protein expression in murine oviductal epithelial cells [135] and human EOC cell lines [136]. Importantly, p53^R273H^ and p53^R248W^ induced distinct levels of FOXM1 protein, suggesting a more complex mechanism than a simple loss of wild-type function [135]. While GOF p53^R175^ and p53^Y220^ are also common ovarian cancer *TP53* mutations, their relationship with FOXM1 expression has not been established [130].

Paired box transcription factor 8 (PAX8) is an FTE lineage marker that is retained during malignant transformation to STICs and HGSC [92] and appears partially responsible for FOXM1 upregulation in HGSC [137]. In cells of the Müllerian (female genital tract) lineage, PAX8 increased *TP53* gene expression regardless of *TP53* mutational status [138]. Since nearly all HGSC cases contain mutant p53, it is plausible that PAX8 upregulates FOXM1 expression in a subset of HGSC by enhancing the expression of GOF p53.

One mechanism by which GOF p53 increases FOXM1 expression in EOC is by enhancing FOXM1 mRNA stability [136]. Other GOF p53 mutations, such as p53^G245D^, have been shown to upregulate *FOXM1* expression in other cancers [139]. Specifically, p53^G245D^ decreased AMP-activated protein kinase (AMPK)-mediated phosphorylation of FOXO3A in head and neck squamous cell carcinoma cells, thereby alleviating its repression on FOXM1 gene expression [139]. Thus, upregulation of FOXM1 expression in ovarian cancer may occur through distinct mechanisms depending on the specific *TP53* mutation.

Inactivation of retinoblastoma (Rb) (*RB1*) by truncating point mutations, indels, and gene breakage occurs in 17.5% of HGSC tumors [116]. Moreover, the Rb pathway is altered in 67% of HGSC tumors, rendering it a very common molecular defect [30]. In addition, the Rb and p53 signaling pathways are highly interconnected [140]. In a murine oviductal line continually passaged in culture to mimic cellular aging, high passage number cells demonstrated molecular changes including increased FOXM1 expression, hyper-phosphorylated (i.e, inactivated) Rb, and the expression of a p53 splice variant that exhibited partial loss of wild-type p53 function with potential GOF [141]. Our studies demonstrated that combined *TP53* and *RB1* inactivation in murine and human OSE cells synergistically upregulates FOXM1 mRNA and protein expression than when compared to inactivation of either tumor suppressor alone [127]. Moreover, we showed that FOXM1 is overexpressed in a transgenic murine model of ovarian cancer driven by combined *Rb1*/*Trp53* knockout in the OSE [127]. Downregulation or inactivation of the p53 and Rb pathways result in the activation of the E2F1 transcription factor, which directly upregulates FOXM1 gene expression by binding to its promoter [133,142,143,144,145]. Indeed, we reported that *E2F1* knockdown in human OSE cells and in the HGSC cell line COV362 with inactivated p53 and Rb significantly reduced FOXM1 mRNA expression [127]. In agreement, *E2F1* mRNA showed a strong positive correlation with FOXM1 mRNA in human primary EOC [127]. We also have shown that dysregulation of the Rb/E2F pathway, including the overexpression of E2F1 or cyclin E1 or *RB1* knockout, caused increased FOXM1 expression in FTE cells [27]. *TP53* mutation, *RB1* copy number loss, and *CCNE1* expression each strongly correlates with FOXM1 expression and FOXM1 pathway activation in pan-cancer [27]. These findings establish that p53 and Rb pathway dysregulation is a key contributor to FOXM1 overexpression in ovarian cancer.

FOXO3A is a FOX family member and tumor suppressor that functions as a negative regulator of FOXM1. FOXO3A suppresses FOXM1 activity via at least three mechanisms: (1) Downregulating FOXM1 gene expression, potentially through the upregulation of the Mad/Max family of transcriptional repressors [146,147,148]; (2) directly displacing FOXM1 from the promoter region of FOXM1 target genes [148,149,150]; and (3) condensing FOXM1 gene targets into heterochromatin, making the genes less accessible to FOXM1-mediated activation [148]. Critically, as normal FTE progresses to STIC lesions and HGSC, FOXO3A expression steadily decreases with a concurrent increase in FOXM1 expression and activation of the PI3K/AKT and MAPK/ERK pathways [151]. Several mechanisms may result in the loss of FOXO3A expression in HGSC. First, late-stage HGSC tumors demonstrated *FOXO3A* copy number loss and a subset of TCGA HGSC showed upregulation of miR-182, which targets *FOXO3A* mRNA for degradation [151]. Second, FOXO3A phosphorylation by the PI3K/AKT or MAPK/ERK pathways, which are highly active in HGSC, results in the nuclear export of FOXO3A to the cytoplasm and thus inactivation of its transcriptional activity [151,152]. Third, T-type Ca^2+^ channels, which are aberrantly overexpressed in ovarian cancer, can activate the PI3K/AKT pathway, thus antagonizing FOXO3A nuclear retention and promoting FOXM1 expression [153]. Consequently, selective inhibitors against PI3K, ERK1/2, and AKT were shown to significantly decrease both basal and induced FOXM1 gene expression levels in HGSC cells [154]. In agreement, inhibiting T-Like Cell-Originated Protein Kinase (TOPK), which is a protein in the MAPK/ERK pathway that also modulates the PI3K/AKT pathway, leads to decreased FOXM1 gene expression in ovarian cancer cells [155].

### 3.3. Upstream Oncogenes Promote FOXM1 Expression

Several oncogenic transcription factors have been noted to promote FOXM1 overexpression in ovarian cancer. Yes-associated protein (YAP) can bind to the FOXM1 promoter and promotes FOXM1 gene expression in HGSC cell lines [154]. Investigations in mesothelioma and soft-tissue sarcoma cell lines have also demonstrated the ability of the YAP-TEAD complex to transcriptionally activate FOXM1 expression [156,157], and there is evidence in soft-tissue sarcoma that FOXM1 can directly interact with the YAP-TEAD complex to promote cell proliferation [157]. In HGSC, bromodomain and extraterminal domain (BET) proteins recognize acetylated lysine residues on histones and may recruit chromatin-modification enzymes or co-activators to activate the FOXM1 promoter [158]. Indeed, a pan-inhibitor of the BET family, I-BET151, downregulated FOXM1 mRNA and protein expression in several EOC cell lines [159]. Given the emergence of BET inhibitors as a novel therapeutic approach in EOC [160], further studies of the effect of these agents on FOXM1 expression and FOXM1-driven phenotypes are warranted. Additional proteins, which are not classical oncogenes, are also known to upregulate FOXM1 expression in ovarian cancer. For example, the ETS family transcription factor ETV5 directly binds to the FOXM1 promoter to upregulate its expression in EOC [161]. Additionally, the E3 ubiquitin ligase substrate receptor VprBP/DCAF1, which is upregulated in HGSC, co-activates FOXM1 expression by a mechanism independent of its E3 ubiquitin ligase activity [47].

Studies in other cancer types have identified well-known oncogenic transcription factors that bind to the FOXM1 promoter and induces its transcription: Gli1 and Gli2 in the sonic hedgehog (Shh) pathway [162,163,164,165], c-Myc [145,166,167,168], STAT3 [169], and Twist1 [170]. c-Myc is known to transcriptionally activate FOXM1 in several cancers [166,167,168] and, notably, exhibits copy number gains in >65% of HGSC cases [30]. In addition, FOXM1 has been reported to bind to its own promoter and induce its own gene expression through an autoregulatory circuit [171,172]. However, our data were unable to confirm this mechanism [27].

Although not addressed in ovarian cancer, there is evidence that stress response proteins increase FOXM1 expression; this suggests that FOXM1 promotes cell survival in harsh conditions. For example, heatshock factor 1 (HSF1) can bind to the FOXM1 promoter and upregulate its expression when glioma cells are subjected to lethal heat shock stress [173]. Activating transcription factor 4 (ATF4), which is a transcriptional regulator that responds to amino acid deprivation, can upregulate histone lysine demethylase KDM4C, which activates transcription of FOXM1 and FOXM1 target genes [174].

Reactive oxygen species (ROS) provide another link between cell stress and FOXM1. ROS can upregulate FOXM1 expression, which then activates transcription of ROS scavenger proteins such as manganese superoxide dismutase (MnSOD) [175]. MnSOD in turn promotes E2F1-mediated and Sp1-mediated activation of the FOXM1 promoter [176]. These findings suggest that ROS activates a feed forward loop that leads to the increased protein expression of MnSOD and FOXM1. Additionally, HIF1α, in the presence of ROS or hypoxia, binds to and transactivates the FOXM1 promoter [177,178]. In the context of ovarian cancer, the follicular fluid microenvironment is a major source of ROS, which plays important physiologic functions in the ovaries, including follicular growth, oocyte maturation, and ovarian steroid biosynthesis [179]. The oxidative stress exerted on the ovaries and FT could potentially result in FOXM1 upregulation in premalignant and malignant ovarian cells, which may in turn drive cellular proliferation in cells that might normally undergo DNA damage arrest.

### 3.4. FOXM1 Is Post-Transcriptionally Regulated by Non-Coding RNAs (ncRNA)

Non-coding RNAs (ncRNA) compose a complex regulatory network that participates in diverse biological processes ranging from chromatin remodeling to protein stabilization [180,181]. In the context of FOXM1, ncRNA investigations have focused on post-transcriptional regulation of the *FOXM1* mRNA. Among the different classes of ncRNAs, microRNAs (miRNA) bind to and destabilize messenger RNAs (mRNA), while long non-coding RNAs (lncRNA) and circular RNAs (circRNA) can contain miRNA-complementary sites that competitively bind to miRNAs and reduce the number of available miRNAs that can bind to mRNA targets; this is an action known as miRNA sponging [180,181]. Thus, ncRNAs play a major role in FOXM1 regulation, with over 40 implicated to date [182].

In ovarian cancer, miR-370 targets *FOXM1* mRNA for degradation [183,184]. This mechanism has also been observed in osteosarcoma [185], gastric cancer [186], and acute myeloid leukemia [187]. CircRNA hsa_circ_0061140 and lncRNA plasmacytoma variant translocation 1 (lnc*PVT1*), for which expressions are upregulated in ovarian cancer, sponge miR-370 activity in ovarian cancer cells [183,184]. Studies in gastric cancer and pan-cancer have suggested that lnc*PVT1* can also interact with and stabilize the FOXM1 protein and have shown that FOXM1 binds to the promoter of lnc*PVT1* to activate its transcription [188,189]. Additionally, lnc*PVT1* can interact with *FOXM1* mRNA directly, potentially regulating *FOXM1* mRNA splicing and stability [189]. *PVT1* also exists as a circRNA (circ*PVT1*) [190]. In EOC, circ*PVT1* is able to sponge miR-149-5p, which targets *FOXM1* mRNA [191]. miR-149 has also been shown to repress FOXM1 expression in non-small-cell lung cancer [192] and gastric cancer [193]. Interestingly, *PVT1* is located on chromosome band 8q24.21 and is frequently co-amplified with *MYC* [30,194]. Additional miRNAs established to target FOXM1 in ovarian cancer are miR-134 [195,196,197,198,199] and miR-216b [200,201,202,203]. Interestingly, a recent report indicates that FOXM1 acts as a transcriptional regulator of several miRNA molecules in triple-negative breast cancer, which has a similar molecular profile to HGSC [204]. Taken together, these data indicate that ncRNA alterations play a key role in promoting FOXM1 expression in cancer, including ovarian cancer, and that FOXM1 itself can function in miRNA dysregulation.

### 3.5. FOXM1 Is Stabilized and Activated by Post-Translational Mechanisms

As mentioned earlier, FOXM1 is functionally regulated by post-translational modifications (PTMs). In particular, a plethora of evidence supports that FOXM1 protein stabilization and transcriptional activity depend on specific phosphorylation events [29,37,49,50,51,52,53,54,55,56,57,59]. However, to date, most FOXM1 expression studies in cancer, including ovarian cancer, have only reported overall FOXM1 protein levels rather than phosphorylated FOXM1 (phospho-FOXM1), which stems from a limited number of antibodies available for PTMs. Measuring phospho-FOXM1 may provide more insight into FOXM1 activation status in cancer and should be emphasized in future investigations. Known FOXM1 phosphorylation sites of kinases known to be relevant in ovarian cancer are illustrated in Figure 2.

The MAPK/ERK and PI3K/AKT pathways promote FOXM1 activation in cancer, including ovarian cancer. As described earlier, the overabundance of T-type Ca^2+^ channels in ovarian cancer leads to PI3K/AKT overexpression and, consequently, treatment with mibefradil, which is a calcium channel blocking agent, decreased nuclear FOXM1 protein levels and its binding to the *BIRC5* (survivin) promoter in EOC cells [153]. Growth factor receptor-bound protein 7 (GRB7), which is a signal transducing adaptor protein that is overexpressed in ovarian cancer, promotes constitutive activation of the MAPK/ERK pathway and thereby leads to enhanced FOXM1 activity [205]. Interestingly, the MEK inhibitor U0126 decreased FOXM1 protein levels in OVCA433, which is an ovarian cancer cell line with functional p53, but not in SKOV3, a p53-deleted ovarian cancer cell line; this suggests that functional p53 may be necessary for the MAPK/ERK pathway to regulate FOXM1 activity [206]. Studies in other cancers have suggested that the PI3K/AKT pathway, independent of the MAPK/ERK pathway, functions as a main activator of oncogenic FOXM1 activity [207,208,209].

Additional kinases also promote FOXM1 phosphorylation in ovarian cancer. HGSC tumors with downregulation of miR-506, which targets *CDK4/6* mRNA, showed increased FOXM1 protein [210]. Since CDK4/6 phosphorylates FOXM1c at numerous residues (Figure 2) [49], this result implies that the gain of CDK4/6 increased FOXM1 protein activity and stability in these tumors. Overexpression of Polo-like kinase 1 (PLK1), for which phosphorylation at S730 at the TAD of FOXM1c (S715 on FOXM1b) relieves the physical repression of the FOXM1 NRD [59], is prognostic for poor overall survival in EOC [211,212]. Elucidating the relative contributions of different kinases to FOXM1 activation in ovarian cancer may provide novel avenues for therapeutic intervention in this disease.

Deubiquitination serves as an additional mechanism to increase FOXM1 expression in ovarian cancer. Otubain 1 (OTUB1) belongs to a family of deubiquitinases (DUBs), for which their defining feature is their ovarian tumor (OTU) domain, and OTUB1 promotes aggressive behavior in several cancers through both canonical and non-canonical DUB functions [213]. In ovarian cancer, OTUB1 promoted ovarian cancer cell proliferation, invasion, and tumor growth through deubiquitination and stabilization of FOXM1 [214]. It is likely that other DUBs may stabilize FOXM1 protein in ovarian cancer. For example, in basal breast cancer, which has robust molecular similarity to HGSC [215], the DUB USP21 deubiquitinates and stabilizes FOXM1 in vitro and in vivo [216]. Further elucidations of DUBs that target FOXM1 in ovarian cancer are an important area for future investigations.

## 4. FOXM1 Oncogenic Functions

FOXM1 is a transcriptional master regulator of several hallmarks of cancer [217,218]. Upregulation and activation of FOXM1 in cancer can contribute to numerous phenotypes, including cell proliferation, cancer stemness, genomic instability, drug resistance, protection from oxidative stress, altered metabolism, invasion, metastasis, angiogenesis, and inflammation [148,217,219,220]. Remarkably, and consistent with its function in a myriad of oncogenic phenotypes, FOXM1 has been reported as the top gene expression biomarker for poor prognosis in a pan-cancer analysis consisting of >18,000 tumors from 39 distinct malignancies [221]. Our present understanding of FOXM1 in relation to ovarian cancer phenotypes is summarized in Figure 4. Mechanistically, FOXM1 activates genes by binding to gene promoters and enhancers, both directly [40] and via interactions with transcription factor complexes such as MuvB [41,42,43], B-Myb [41,42,43], and NFY [44]. FOXM1 transcriptional targets identified in ovarian cancer are shown in Table 2.

### 4.1. FOXM1 Expression Is Associated with Tumor Progression and Poor Prognosis in Ovarian Cancer

FOXM1 expression is elevated in multiple stages of ovarian cancer, from initial neoplastic transformation to late-stage metastatic spread. In one study, immortalized FT stem cells were observed to have higher *FOXM1* expression than non-immortalized counterparts and STICs isolated from women with HGSC revealed higher expression of *FOXM1* than normal FTE [236]. In addition, FOXM1 is expressed in STIC lesions in concert with FOXO3A downregulation [151]. These data suggest the FOXM1 upregulation in HGSC may occur early and prior to full neoplastic transformation. However, FOXM1 expression clearly shows additional elevation in later disease stages. For example, *FOXM1* gene expression in primary EOC, including HGSC, is highly overexpressed compared to normal epithelial ovarian tissues [30,127,134,161,222] and directly correlates with the tumor stages [127] and grade [127,161]. FOXM1 protein expression in EOC positively correlates with lymph node metastasis [237] and FIGO stage [222,238]. We reported that HGSC TCGA cases with *FOXM1* gene amplification have overall reduced survival [127] and other studies have demonstrated that FOXM1 protein expression directly correlates with reduced disease-free [239], progression-free [237,240], and overall [237,239] survival in EOC.

### 4.2. FOXM1 Promotes Cellular Proliferation, Migration, and Invasion

Increased FOXM1 expression in cancer promotes cell cycle progression and cell proliferation. In particular, FOXM1 regulation of cell cycle and mitotic genes, such as *CCNB1*, *CDK1*, and *CENFP*, is conserved across different cancer types [44]. We observed that FOXM1 knockdown in immortalized human OSE cells results in the accumulation of cells in G2/M, while FOXM1 knockdown in HGSC cells led to accumulation of cells in G1 [127]. In both cell models, FOXM1 knockdown downregulated the expression of *SKP2*, *PLK1*, and *CCNB1* and this is consistent with inhibition of cell cycle transitions [127].

Additional studies have demonstrated that FOXM1 promotes ovarian cancer cell migration and invasion. Stabilization of FOXM1 via deubiquitination by OTUB1 promotes cell proliferation and invasion of SKOV3 cells in vitro and increased tumor growth of SKOV3 mouse xenografts in vivo [214]. A separate study demonstrated that FOXM1 knockdown in SKOV3 cells inhibited cell migration and invasion [239]. Conversely, FOXM1 overexpression in A2780/CP70 and OVCA433 cells increased proliferation, migration, and invasion. Mechanistically, in OVCA433 cells, FOXM1 overexpression increased mRNA expression of the extracellular proteases *MMP9* and *PLAUR* [206], while the FOXM1 inhibitor thiostrepton decreased cell migration and invasion in conjunction with reducing *MMP9* and *PLAUR* expression [206]. In another study, FOXM1 knockdown decreased ovarian cancer tumor growth in xenografts [229]. FOXM1 knockdown in the EOC cell lines EOC-CC1 and OSPC2 inhibited cell proliferation, colony formation, and invasion, which coincided with decreased expression cell cycle genes and metastasis genes [222].

FOXM1 gene targets that are relevant to cell proliferation, migration, and invasion have been described. Notably, many are overexpressed in ovarian cancer and are reported to correlate with poor prognosis. These include: *SKP2* [127], *PLK1* [127], *CCNB1* [127,222], *CDC25B* [222], *CEP55* [222], *CENPF* [222], *TOP2A* [222], *CCNF* [223], *PRC1* [223], and *MMP2* [222]. In addition, several novel FOXM1 gene targets have been identified in ovarian cancer. For example, the cytoskeleton proteins cytokeratin-5 (*KRT5*) and cytokeratin-7 (*KRT7*) are known to promote migration in SKOV3 cells [226]. *DLX1* [224], *PCLAF* [225], *KIF20A* [223], and *CTNNB1* [229] have been shown to contribute to proliferation and metastatic phenotypes. Moreover, FOXM1 target genes can be linked to broader cancer pathways. KRT5 is found in a population of basally located CD44^+^ stem-like cells in the FTE, for which its population is expanded in serous cancer samples [227]. KRT7 upregulates integrin β1-FAK signaling and matrix metalloproteinase expression, promoting cell matrix adhesion [241] and extracellular matrix degradation [228], respectively. The homeobox protein DLX-1 is a mediator in TGF-β1/SMAD4 signaling in ovarian cancer [224], which promotes EMT and cell stemness phenotypes [242]. The proliferation cell nuclear antigen (PCNA) clamp-associated factor PCLAF activates the PI3K/AKT/mTOR pathway in ovarian cancer cells [225]. *CTNBB1* encodes β-catenin, which contributes to ovarian cancer metastasis, stemness, chemoresistance, angiogenesis, and immune evasion [230]. Interestingly, FOXM1 can bind directly to β-catenin to promote its nuclear localization and transcriptional activity in glioma; this provides an addition link between FOXM1 and β-catenin [243].

In ovarian cancer, FOXM1 also induces the expression of integrins and matrix proteins: *ITGB1* [231], *ITGAV* [231], *ITGA5* [231], *LMNB1* [231], and *FN1* [231], which may facilitate adhesion of ovarian cancer cells to new organs. In keratinocytes, ectopic overexpression of FOXM1 in combination with the loss of *TP53* enhanced integrin β1 expression [145]. Consistently, the lung vasculature in *FOXM1*^−*/*−^ mouse embryos demonstrated a downregulation of integrin β1 and laminins α2 and α4, and FOXM1 was demonstrated to transactivate the *LAMA4* promoter [21]. There is also evidence that FOXM1 targets E-cadherin, although this has not been reported in ovarian cancer [244]. E-cadherin plays a dynamic role in ovarian cancer, where its overexpression is important for the growth and survival of ovarian cancer cells [245,246], its fragmentation is important for intraperitoneal metastasis [246,247], and its re-expression is a key part of mesenchymal-to-epithelial transition (MET) [245,248]. Taken together, these studies suggest that FOXM1 regulates the expression of key adhesion molecules and promotes their expression in contexts favorable to ovarian cancer progression.

In addition to cancer cells, the tumor microenvironment (TME) plays a critical role in ovarian cancer [249]. While not yet explored in ovarian cancer, investigations in other cancers have shed light on the role of FOXM1 in TME development and maintenance. A mouse model with a macrophage-specific *Foxm1* deletion (mac*Foxm1*^−/−^) demonstrated decreased macrophage recruitment and migration to lung tumors and ultimately reduced the number and size of lung tumors formed [250]. In cancer-associated fibroblasts (CAFs), FOXM1 and its downstream targets are upregulated in several cancers [251] and hepatocellular carcinoma CAFs rely on FOXM1 to activate cartilage oligomeric matrix protein (*COMP*) gene expression, which eventually increases EMT, invasion, and stemness of hepatocellular carcinoma cells [252]. The recent development of mouse models that recapitulate TME observed in human ovarian cancer [253,254] provides a ripe opportunity to study the role of FOXM1 in the ovarian cancer TME.

### 4.3. FOXM1 Promotes DNA Repair and Chemotherapy Resistance

Most ovarian cancers are diagnosed at advanced stages and ultimately develop chemoresistance. FOXM1, whose transcriptional network includes DNA repair genes [255], has been reported to promote ovarian cancer resistance to taxanes (e.g., paclitaxel and docetaxel), platinum-based drugs (e.g., cisplatin and carboplatin), and PARPi (e.g., olaparib and niraparib). For example, cisplatin-resistant or paclitaxel-resistant IGROV1 cells showed significant increases in FOXM1 expression, and FOXM1 contributed to the chemoresistant phenotype [240]. Additionally, FOXM1 knockdown in EOC cell lines EOC-CC1 and OSPC2 decreased the expression of DNA damage response genes (*ASPM*, *XRCC1*, *XRCC4*, and *RAD51*) and chemoresistance genes (*CXCR4*, *CYR61*, and *EDN1*) and concomitantly increased cell sensitivity to several chemotherapy agents, including carboplatin, cisplatin, doxorubicin, and olaparib [222].

In HGSC patients, shallow whole genome sequencing (sWGS) of circulating tumor DNA (ctDNA) from plasma revealed a 16% increase in chromosome 12p13.33 amplification (location of *FOXM1*) after the acquisition of chemotherapy resistance [128]. In agreement, we observed that almost half of HGSC patients have increased FOXM1 expression in recurrent chemoresistant tumors [256]. Bioinformatic analysis led to the identification of FOXM1 as one of the top three hub genes for which overexpression leads to platinum-based chemotherapy resistance [257]. Furthermore, cisplatin-resistant ovarian cancer tissues and cells were reported to have increased FOXM1 [232] and FOXM1 was an independent indicator of shorter time to progression in platinum-resistant EOC [222]. A2780 and SKOV3 cells treated with cisplatin demonstrated increased FOXM1 protein expression in a dose-dependent manner [232]. Additionally, ectopic overexpression in BG-1 and A2780 cells enhanced cisplatin resistance, while FOXM1 knockdown sensitized SKOV3 and A2780/CP70 cells to cisplatin [232,240]. Taken together, these data implicate FOXM1 in promoting chemoresistance, particularly to platinum-based drugs, in ovarian cancer.

Several studies have reported that the FOXM1 small molecule inhibitor (SMI) thiostrepton sensitizes ovarian cancer cells to platinum-based chemotherapy [258,259]. For example, thiostrepton in combination with cisplatin increased cell death in ovarian cancer cell lines and human ovarian cancer ascites cells ex vivo than compared to treatment with platinum alone [260]. In EOC cell lines A2780 and HEC-1A, treatment with thiostrepton (1–10 µM) in combination with cisplatin had a synergistic effect on cell death [136]. Pretreatment of the cisplatin-resistant ovarian cancer cell line A2780/CP70 with thiostrepton increased cisplatin sensitivity in vitro and in xenografts [240]. However, the mechanistic link between these phenotypes and FOXM1 is uncertain since thiostrepton has pleiotropic effects [261,262,263].

Investigations of FOXM1 also support that there is a role with respect to PARPi and taxane resistance in ovarian cancer [222,233,240]. For example, treatment of ovarian cancer patient tissues ex vivo with the PARPi olaparib increased *BRCA1*, *BRCA2*, *RAD51*, and FOXM1 gene expression, and treatment of the tissues with olaparib and thiostrepton reversed this effect. Moreover, the combination treatment, but not olaparib alone, led to decreased proliferation and increased apoptosis [234]. Furthermore, FOXM1 inhibition by siRNA or thiostrepton sensitized EOC cells to olaparib, which correlated with apoptosis, increased DNA damage, and increased PARP1 trapping [233]. Thiostrepton treatment sensitized rucaparib-resistant EOC cell lines to the PARPi rucaparib [233]. FOXM1 has also been shown to promote taxane resistance in breast cancer [216,264,265], gastric cancer [266,267,268], hepatocellular carcinoma [269], prostate cancer [270], and nasopharyngeal carcinoma [271].

FOXM1 transcriptional targets linked to platinum resistance in ovarian cancer include exonuclease 1 (EXO1) [232], a DNA repair protein recruited to double-stranded breaks, and PCLAF [225], a protein that activates the PI3K/AKT/mTOR signaling pathway. In addition, Protein regulator of cytokinesis-1 (PRC1), which promotes proliferation, migration, and invasion, is a direct FOXM1 transcriptional target found to promote resistance to multiple agents (cisplatin, taxol, and doxorubicin) [223]. Finally, olaparib treatment of EOC cells stimulated FOXM1 binding to the promoters of *CCNB1* [233], *BRCA1* [233,234], *BRCA2.* [234], *RAD51* [233,234], *FANCF* [233], *RAD51D* [233], and *FANCD2* [233], which, given the function of these genes, are likely to promote PARPi resistance. FOXM1 is also associated with the upregulation of inhibitor of apoptosis (IAP) genes, including survivin (*BIRC5*) and X chromosome-linked IAP (*XIAP*) in several cancers [272,273,274,275]. Further research on the link between FOXM1 and IAPs, which are intimately related to anti-cancer therapy resistance [276], may reveal additional FOXM1 transcriptional targets linked to ovarian cancer chemoresistance.

### 4.4. FOXM1 Promotes Cancer Cell Stemness

Cancer stem cells (CSC) have enhanced capacities for self-renewal, cell plasticity, and the ability to adapt to harsh environments [277,278,279]. These characteristics facilitate therapy resistance, disease recurrence, and reduced patient survival [277,278,279]. As ovarian cancer has high recurrence rates, targeting stemness is of particular interest [278,279]. The FOXM1 transcriptional network includes pluripotency genes, such as *SOX2*, *NANOG*, and *OCT4*, in several cancer models [280,281,282,283]. Interestingly, FOXM1 depletion in human embryonic stem cells led to a disruption in proliferation but did not impact *OCT4* and *NANOG* expression during in vitro differentiation [235], suggesting that the ability of FOXM1 to modulate pluripotency may be restricted to specific settings which cancer cells are able to exploit.

In ovarian cancer, FOXM1 has been reported to promote cancer cell stemness. CSCs generated from EOC cell lines SKOV3 and A2780 demonstrated elevated levels of FOXM1 in addition to the CSC markers CD133, CD44, and ALDH1 [284]. The synthetic genistein analogue 7-difluormethoxyl-5,4′-di-n-octylgenistein (DFOG) downregulated FOXM1 expression concurrent with CD133, CD44, and ALDH1 and dramatically attenuated sphere-forming abilities [284]. In agreement, ectopic expression of FOXM1 reversed these effects [284]. ALDH1-high cells isolated from A2780 and the cisplatin-resistant sub-line A2780/CP70 expressed increased levels of FOXM1, NOTCH1, OCT4, and NANOG and demonstrated enhanced sphere-forming abilities [285]. Treatment of these cells with the ALDH1 inhibitor diethylaminobenzaldehyde (DEAB) downregulated sphere-forming abilities and FOXM1 expression, while thiostrepton treatment did not affect ALDH1 expression, suggesting that FOXM1 is downstream of ADLH1-induced stemness [285]. Chemoresistance can also be a sign of stemness. Indeed, the cisplatin-resistant A2780/CP70 cell line demonstrated enhanced sphere formation ability and increased protein levels of FOXM1, ALDH1, OCT4, NANOG, and NOTCH1 compared to the A2780 parental line [240]. FOXM1 overexpression in A2780 and BG-1 cells increased sphere formation, while knockdown of FOXM1 in A2780/CP70 and SKOV3 cells decreased sphere formation [240]. In addition to ovarian cancer, FOXM1 is linked to CSC phenotypes in other cancers, including lung [286,287,288], liver [289,290], glioma [291,292,293], breast [294], colon [295], prostate [270], endometrial [296], and embryonal carcinoma [297].

### 4.5. FOXM1 Promotes Genomic Instability and DNA Replication Stress

A defining feature of HGSC is genomic instability, manifested in large part by increased SCNAs [30,116,298]. A seminal early study showed that FOXM1 and several of its transcriptional targets, such as *AURKB* and *CCNB1*, rank in the top 70 genes for which overexpression is associated with chromosomal instability (CIN) in pan-cancer (named the CIN70 signature) [299]. More recently, we and others have reported that FOXM1 and its transcriptional program are enriched in tumors with elevated CIN/functional aneuploidy [27,299,300]. Interestingly, we have also observed significant association between FOXM1, genomic instability, and DNA hypomethylation in EOC [301]. There is a well-established link between DNA hypomethylation and genomic instability in cancer [302,303]. Intriguingly, ectopic FOXM1 expression in oral keratinocytes was shown to concurrently induce genomic instability and DNA hypomethylation [304]. The pattern of DNA hypomethylation in these cells resembled that seen in the head and neck squamous cell carcinoma cell line SCC15 [305]. It is unknown whether FOXM1 can similarly induce genomic instability and DNA hypomethylation in ovarian cancer cell models.

In epidermal keratinocytes, ectopic FOXM1 expression and UVB exposure induced SCNAs and loss of heterozygosity (LOH), predisposing cells to a “second hit” on the DNA-damage checkpoint responses (e.g, *TP53* mutations) that promote malignant transformation [306]. Moreover, FOXM1 promoted the proliferation and attenuated the differentiation of keratinocytes [145]. These observations suggest that FOXM1 may drive accelerated G2/M progression, thus preventing cells with irreparable DNA damage from committing to terminal differentiation [145].

DNA replication stress (RS) is caused by extrinsic and intrinsic factors that disrupt replication fork dynamics, and critically, RS is a major driver of cancer genomic instability [307,308]. Notably, FOXM1 was recently reported to induce DNA replication stress in vitro and FOXM1 expression was observed to correlate with expression of RS biomarkers in several cancer types, including HGSC [309]. Our recent study also suggests a link between FOXM1 and RS. We showed that *RHNO1*, which encodes a DNA damage repair protein involved in the cellular RS response, shares a head-to-head (i.e., bidirectional) gene arrangement with FOXM1 on chromosome 12p13.33 [256]. Activation of the FOXM1/RHNO1 bidirectional promoter (F/R-BDP) leads to balanced gene expression of both genes and we observed that FOXM1 and RHNO1 each promote HGSC cell growth, survival, and homologous recombination (HR) DNA damage repair [256]. Importantly, FOXM1 and RHNO1 promoted olaparib and carboplatin resistance, and CRISPR-dCas9-mediated repression of the F/R-BDP reversed these effects [256]. We postulate that cancer cells have a selective advantage for FOXM1 and RHNO1 co-expression since FOXM1 drives RS and downstream oncogenic phenotypes such as genomic instability, while RHNO1 helps mitigate FOXM1-induced RS to a level more favorable to cancer cell survival.

### 4.6. FOXM1 Is Linked to Altered Cellular Metabolism

FOXM1 is known to contribute to metabolic cellular processes. For example, a proteomics study of the breast cancer cell line MCF-7 revealed that FOXM1 altered the expression of 37 proteins associated with mitochondrial biogenesis and glycolysis [294]. In pancreatic cancer, FOXM1 expression was upregulated in a glucose-dependent manner, which correlated with epithelial-to-mesenchymal transition (EMT) in pancreatic cancer cell lines [310]. In addition, downregulation of FOXM1 induced pancreatic cancer cells to use mitochondrial respiration rather than aerobic glycolysis in high-glucose medium, further linking FOXM1 to the glycolytic pathway [310]. Similarly, FOXM1 knockdown decreased glucose utilization, lactate production, and lactate dehydrogenase (LDH) activity in several pancreatic cell lines and downregulated phosphoglycerate kinase 1 (PGK-1) and lactate dehydrogenase A (LDHA) [311]. Further investigation revealed that FOXM1 binds directly to the *LDHA* promoter to promote its expression in pancreatic cancer cells [311]. FOXM1 was also found to transactivate *LDHA* in gastric cancer, promoting a glycolytic phenotype with high proliferative, migratory, and invasive abilities [312]. Similarly, FOXM1 knockdown decreased glucose uptake and lactate production in the hepatocellular carcinoma cell line Hep3B, and the overexpression of FOXM1 in the hepatocellular carcinoma cell line MHCC-97H increased glucose uptake and lactate production, but only when appropriate levels of glucose transporter 1 (GLUT1) were present [313]. As part of this regulatory mechanism, FOXM1 directly transactivated the *GLUT1* promoter [313]. FOXM1 has also been shown to regulate the level of the oncometabolite D-2-hydroxyglutarate, by activating the isocitrate dehydrogenase 1 (*IDH1*) promoter in the fibrosarcoma cell line HT-1080 [314].

Two observations support that FOXM1 may also be important for metabolic reprogramming in ovarian cancer. First, *FOXM1* gene expression is upregulated in ovarian cancer cells by both the metabolic enzyme aspartate N-acetyltransferase, which is overexpressed and associated with worse clinical outcomes in HGSC, and N-acetylaspartate, which is the most abundant onco-metabolite in HGSC tissues [315]. Second, FOXM1 directly binds to the promoters of *GLUT1* and *hexokinase 2* (*HK2*) in EOC cell lines A2780 and SKOV3, and FOXM1 knockdown resulted in decreased aerobic glycolysis in these cell lines [316]. Moreover, both mRNA and protein expression of GLUT1 and HK2 positively correlated with FOXM1 in EOC tissues [235]. Taken together, these data suggest that altered amino acid metabolism in ovarian cancer cells upregulates FOXM1, which alters glycolysis and mitochondrial respiration by promoting aerobic glycolysis. The Warburg effect, wherein tumor cells metabolize glucose through aerobic glycolysis as opposed to oxidative phosphorylation, is well-known for promoting proliferation, metastasis, stemness, and therapy resistance [317,318]. Furthermore, as glycolysis and mitochondrial activity affect fatty acid metabolism [319], FOXM1 may also play a role in fatty acid metabolism. Further study on the relationship between FOXM1 and ovarian cancer metabolism is highly encouraged.

### 4.7. FOXM1 Isoform Expression and Function in Cancer

As mentioned earlier, FOXM1 is distinguished by three principal isoforms (a, b, and c) (Figure 1). While several early cancer investigations focused on *FOXM1b* [32,320], our recent studies of the TCGA database and human HGSC tissues revealed that *FOXM1c* is the highest expressed FOXM1 isoform in both pan-cancer and HGSC [27,127]. Consistent with our findings, it was demonstrated that *FOXM1c* had the highest expression compared to *FOXM1b* and *FOXM1a* in a panel of EOC cell lines [321]. *FOXM1c* has also been associated with proteolytic processing that removes the NRD, which may result in constitutive activation [322]. In a bioinformatic study analyzing FOXM1 isoforms enriched in 12 ovarian cancer tissues compared to 18 normal control tissues, two novel FOXM1 isoforms missing the NRD were identified within the top 5% enriched isoform genes [36]. However, the sequences of these isoforms suggested that they are not translated into proteins and their functional contribution to ovarian cancer remains unknown (Table 1).

It is likely that FOXM1b and FOXM1c have common transcriptional targets, given the similarities of the proteins as well as data showing that both isoforms can target oncogenic *DLX1* in ovarian cancer cells [224]. In addition, our recent isoform-specific overexpression and RNA-sequencing (RNA-seq) study revealed a number of common transcriptional targets between FOXM1b and FOXM1c, as well as some distinct targets [27]. Predictably, gene expression changes were largely absent from FOXM1a overexpressing cells [27] and this is consistent with its reported transcriptional incompetence [12].

Some studies suggest that FOXM1b and FOXM1c expression may have distinct phenotypic consequences. Previous work in several cancer cell lines, including the ovarian cancer cell line A2780/CP70, observed that FOXM1b had a higher transforming ability than FOXM1c and this is as measured by anchorage-independent growth [322]. However, another study reported that FOXM1c promoted proliferation, migration, and invasion, while FOXM1b only promoted cell migration and invasion in EOC cells [206]. In a study of the newly EOC-derived cell lines EOC-CC1 and OSPC2, higher amounts of *FOXM1c* than *FOXM1b* expression were detected (which matched their original clinical biopsy specimens) [222]. In this study, the highest amount of *FOXM1c* compared to *FOXM1b* expression was found in OSPC2 cells from patient ascites [222]; this suggests that FOXM1c may be upregulated when cells have adopted a mesenchymal (metastatic) phenotype. Therefore, while FOXM1b and FOXM1c are both able to promote oncogenic phenotypes, FOXM1c may promote more proliferative and metastatic phenotypes and, due to its higher relative expression [27,127], FOXM1c may be the most relevant isoform to overexpress and model in ovarian cancer investigations.

Some work has specifically linked FOXM1c to the MAPK/ERK pathway. Only FOXM1c, which contains two functional ERK1/2 target sequences [56], was found to be sensitive to activation by MAPK/ERK in HEK293 cells [322]. Similarly, in the mouse fibroblast line NIH/3T3, constitutively active MEK1 promoted FOXM1c, but not FOXM1b, transactivation of the *CCNB1* promoter [56]. As mentioned earlier, the connection between MAPK/ERK and FOXM1 in ovarian cancer has only been demonstrated in EOC cell lines with wild-type p53, and it has been suggested that the MAPK/ERK pathway may not interact with FOXM1 in mutant p53 settings [206]. Therefore, it is not entirely clear if the interaction between MAPK/ERK and FOXM1c is relevant in ovarian cancer.

## 5. Clinical Translation

### 5.1. FOXM1 Has Potential as a Prognostic Biomarker in Ovarian Cancer

Numerous studies have linked FOXM1 expression to poor prognosis in ovarian cancer. *FOXM1* mRNA expression associates with higher EOC tumor grades [127,161] and stages [127], and FOXM1 protein expression associates with EOC lymph node metastasis [237] and a higher FIGO stage [222]. HGSC tumors with *FOXM1* gene amplification have increased *FOXM1* mRNA expression and reduced overall survival [127]. EOC tumors with high *FOXM1* mRNA expression showed reduced progression-free and overall survival [238]. EOC tumors with elevated FOXM1 protein expression were associated with reduced disease-free [239], progression-free [237,240], and overall [237,239] survival. Meta-analyses of FOXM1 in human solid tumors reported that FOXM1 protein expression in ovarian tumors coincides with an overall hazard ratio (HR) of 1.34 (95% CI = 0.96–1.88) for overall survival [323] and an odds ratio (OR) of 2.34 (95% CI = 1.30–4.20) for 3-year overall survival [324]. In addition, during the early stage, non-serous EOC FOXM1 protein levels correlated with poor prognosis in mucinous OC and improved the predictive power of current clinical markers (age, stage, CA-125, and ploidy) [325]. Most remarkably, a pan-cancer meta-analysis of the transcriptomes of ~18,000 human tumors identified FOXM1 expression as the top single gene predictor of poor prognosis in cancer [221]. FOXM1 was also shown to outperform the widespread clinical marker, *MKI67* (encodes Ki-67), for predicting survival [221]. These data suggest FOXM1 should be aggressively pursued as a prognostic ovarian cancer biomarker in clinical validation studies.

### 5.2. In Vivo Studies of FOXM1 in Ovarian Cancer Are Limited

Although in vitro ovarian cancer models provide invaluable insight into the function of FOXM1 in ovarian cancer, cell lines do not adequately recapitulate in vivo disease. For example, cell lines may experience selective pressure, genetic drift, and genomic instability, resulting in phenotypic changes (including drug response) that no longer reflect the original tumor [326,327]. Furthermore, cell lines do not interact with the tumor microenvironment (TME) or engage in metastatic processes in the manner experienced by tumors in vivo. In vivo ovarian cancer models include cell line xenografts, patient-derived xenografts (PDX), syngeneic transplant (i.e., allograft) models, and genetically engineered mouse models (GEMM) [328].

Several considerations drive the need for the continued development of in vivo ovarian cancer models. First, the anti-VEGF monoclonal antibody bevacizumab improves progression-free survival in women with ovarian cancer [329,330], demonstrating the importance of angiogenesis in ovarian cancer. Second, the matrisome [331], including collagen-remodeling genes [332] and cancer-associated fibroblasts (CAF) [333], participates in ovarian cancer progression. Third, while ovarian cancers exist in a generally immunosuppressive environment [334], ovarian tumors contain tumor-associated lymphocytes [221,335,336] and a subset of patients with ovarian cancer respond to immunotherapy [334,337]. Fourth, a key route of ovarian cancer metastasis is via the peritoneal fluid, which carries exfoliated tumor cells to locations including the omentum, peritoneal lining, colon, diaphragm, and small bowel [338]. These metastatic sites can provide different environmental niches for EOC cells. For instance, when metastasizing to the omentum, cancer cells preferentially attach to areas of immune cell aggregates that contain high vascular density [339]. Greater than 70% of ovarian cancer patients have diffuse peritoneal carcinomatosis at initial presentation [338], rendering it a critical process to model in scientific investigations.

The scope of the literature examining the function of FOXM1 using in vivo ovarian cancer models is currently limited. FOXM1 is overexpressed in an ovarian cancer GEMM driven by dual p53/Rb knockout in the OSE [127,340]. Recently, novel allograft models [253,254] and GEMMs [94] were shown to recapitulate critical aspects of human HGSC, including TME and metastasis. Notably, in the GEMM study, *Dicer1-Pten* double knockout (DKO) mice (*Dicer1*^flox/flox^ *Pten*^flox/flox^ *Amhr2*^cre/+^) and *Dicer1-Pten-Trp53* triple knockout (TKO) mice (*p53*^LSL-R172H/+^ *Dicer1*^flox/flox^ *Pten*^flox/flox^ *Amhr2*^cre/+^) were shown to develop tumors exhibiting an activated FOXM1 network, which correlated with genomic instability [94]. In addition to *Tp53*, alterations in DICER1 and PTEN are common alterations in human HGSC and PTEN deletion is linked to FOXO3A downregulation [94,341]. The data from these GEMM models are in agreement with our pan-cancer analysis, which linked FOXM1 mRNA and protein expression to genomic instability [27]. Thus, novel GEMMs may provide highly relevant in vivo models to interrogate FOXM1 function in ovarian cancer development and progression.

### 5.3. Therapeutic Targeting of FOXM1 in Ovarian Cancer

There is strong rationale for targeting FOXM1 in cancer, particularly in aggressive cancers with poor survival outcomes such as ovarian cancer. In general, two therapeutic strategies can be used to impair FOXM1: (1) inhibiting upstream pathways that induce and/or activate FOXM1and (2) inhibiting FOXM1 directly. For the former, several inhibitors of pathways upstream of FOXM1 are used in the clinic or are in clinical trials. In contrast, FOXM1 inhibitors (FOXM1i) have not yet entered clinical trials. However, several direct inhibitors have been used in pre-clinical studies, and there will likely be future clinical testing on these agents. Since ovarian cancers ultimately develop resistance to most chemotherapy, it is worthwhile to develop both indirect and direct FOXM1 inhibitors in parallel.

### 5.4. Inhibitors of Upstream Signaling Pathways

The ErbB family of receptor tyrosine kinases (RTKs) activate the MAPK/ERK [342], PI3K/AKT [342], and PLK1 [146,343] signaling pathways, all of which are upstream kinases that phosphorylate and activate FOXM1. The pan-ErbB receptor inhibitor dacomitinib mitigated FOXM1 activity through reduced levels of phospho-PLK1 in chemotherapy-resistant EOC cells, while single-targeted ErbB inhibitors, such as trastuzumab, had marginal effects on PLK1 and FOXM1 activity [344]. Dacomitinib was also found to reduce FOXM1 activity in pancreatic ductal adenocarcinoma cancer (PDAC) [345], which is another aggressive cancer with poor prognosis and high FOXM1 activity [346]. Selective inhibitors against PI3K, ERK1/2, and AKT decreased FOXM1 gene expression in HGSC cells [154], potentially via their effects on FOXO3A.

PLK1 is a FOXM1 target gene [41] as well as a critical upstream kinase that promotes FOXM1 activation [57,59]. Thus, PLK1 inhibitors (PLK1i) may be a highly effective means to target FOXM1 function. Interestingly, the PLK1i BI6727, combined with paclitaxel, was synthetically lethal in *CCNE1*-amplified HGSC cell lines and triggered mitotic arrest and apoptosis [347]. In combination with dacomitinib, the PLK1i BI2536 synergistically enhanced the sensitivity of chemoresistant EOC cells to cisplatin [344]. It is plausible that the activity of PLK1i observed in these studies involved the disruption of FOXM1. Although BI2536 showed overall limited activity in a phase II trial that included multiple tumor types, the highest proportion of stable disease responses (76.9%) was observed in ovarian cancer patients [348]. Another phase I/II trial, using a *PLK1*-targeted RNAi (TKM-080301), suggested particular efficacy in tumors that overexpress PLK1 and possess an inactivation of wild-type p53 [349], potentially making PLK1i particularly suitable for HGSC. Given their numerous molecular links, it is likely that disruption of FOXM1 signaling accounts for, at least in part, the activity of PLK1i in cancer.

### 5.5. Direct FOXM1 Inhibitors

A variety of molecules that target FOXM1 have now been reported. The tumor suppressor protein p19^ARF^, which is encoded by the *INKA4/ARF* gene locus, inhibits FOXM1 and led to the development of the (D-Arg)_9_-p19^ARF^ 26–44 peptide as a FOXM1 inhibitor [90,91]. Interestingly, this inhibitor targets FOXM1 to the nucleolus, resulting in its inactivation [90]. Nevertheless, potential immune responses to the peptide limits its clinical utility [350]. The thiazole antibiotics thiostrepton [171,259,351] and siomycin A [352] were the first small molecule inhibitors (SMI) reported to inhibit FOXM1, and thiostrepton is the most widely used FOXM1i to date [136,205,206,273,275,353]. One proposed mechanism of action (MOA) of these compounds is that they function as proteosome inhibitors to prevent the degradation of a negative regulator of FOXM1 [354]. Realizing that thiazole antibiotics function as proteosome inhibitors led to the discovery that proteosome inhibitors, such as bortezomib, also inhibit FOXM1 [261,355]. In contrast to this model, another report showed that thiostrepton can bind directly to FOXM1 and suggested that its function as a proteosome inhibitor is a separate effect [351]. More recently, monensin was reported as another antibiotic that inhibits FOXM1 by binding to its DBD [356]. While these agents can have potent anti-cancer effects, their link to FOXM1 is tenuous due to their pleiotropic effects. For instance, thiostrepton inhibits PAX8 in HGSC by a mechanism not dependent on FOXM1 [262] and disrupts mitochondrial protein synthesis [263]. Furthermore, monensin induces mitochondrial ROS production and disrupts Ca^2+^ homeostasis in human cells [357]. Genistein, an isoflavanoid with broad anti-cancer effects [358], has also been proposed as a FOXM1i [359]. In order to increase the bioavailability of genistein, 7-difluoromethoxyl-5,4′-di-n-octylygenistein (DFOG) was synthesized and this compound was shown to downregulate FOXM1 expression in EOC [284,360] and gastric cancer [361] cells.

Several high-throughput screens (HTS) have been performed in order to identify novel SMI of FOXM1. In the first study, the forkhead domain inhibitor FDI-6 was identified from a screen of >50,000 drug-like molecules [362]. FDI-6 disrupted the binding of FOXM1-DBD to RYAAAYA promoter sequences [362]. Although the data presented suggested some specificity to FOXM1, the general mechanism of action of FDI-6 suggests that it might inhibit DNA binding of other FOX family members [362]. Similar to FDI-6, a single-stranded DNA aptamer was designed to target the DBD region of FOXM1, inhibiting FOXM1 transcriptional activity [363]. However, DBD-based inhibitors may not impact the interaction of FOXM1 with other proteins, including oncogenic transcription factor complexes. Thus, inhibitors that result in FOXM1 protein degradation might provide a better therapeutic strategy for targeting FOXM1. In this context, a recent screen of 50,000 small-molecule compounds identified RCM-1 as a FOXM1i [350]. RCM-1 was reported to decrease nuclear FOXM1 protein levels in U2OS C3 cells, and the MOA was reported to involve translocation of nuclear FOXM1 into the cytoplasm, resulting in proteasomal degradation [350]. Another study used computational modeling to screen for FOXM1 SMI in the NCI diversity set of ~2000 synthetic molecules [321]. This strategy identified *N*-phenylphenanthren-9-amine as a molecule that may act similarly to thiostrepton in its binding to FOXM1 [321]. In follow-up work, this compound was shown to inhibit FOXM1 in EOC cells [321]. In another recent study, the 1,1-diarylethylene monoamine compound NB-55 emerged from a chemical library screen as a potent SMI of FOXM1 [364]. This agent was shown to inhibit breast cancer cell proliferation more potently than the proliferation of non-malignant mammary epithelial cells [364]. Using NB-55 as a template, the 1,1-diarylethylene methiodide salts NB-73 and NB-115 were synthesized and shown to have increased potency, with IC_50_ values of ~0.6 µM for proliferation inhibition and reduction in FOXM1 protein [364]. These compounds appear to bind directly to and destabilize FOXM1, resulting in enhanced proteolysis [364]. A synopsis of the anti-cancer effects of direct FOXM1i in ovarian cancer studies is presented in Table 3.

## 6. Conclusions and Future Perspectives

Emerging evidence implicates FOXM1 as a crucial oncoprotein and driver of ovarian cancer. High FOXM1 expression and activity in ovarian cancer are promoted by several mechanisms, including inactivation of upstream tumor suppressors, gene amplification, transcriptional and translational upregulation, increased protein phosphorylation, and enhanced protein stability (Figure 3). In turn, FOXM1 promotes ovarian cancer by impinging on several cancer hallmarks: sustained proliferative signaling, invasion and metastasis, DNA repair and chemotherapy resistance, cancer stemness, DNA replication stress and genomic instability, and altered cell metabolism (Figure 4).

The major gap in our current knowledge of FOXM1 in ovarian cancer is due to the limited number of studies using in vivo or ex vivo models [94,234]. Future work should focus on the verification of the oncogenic potential of FOXM1 using such models, which is required to provide validation for FOXM1 as a therapeutic target in ovarian cancer. Another key opportunity will be to evaluate the impact of FOXM1 status on the efficacy of existing ovarian cancer therapies. Extensive in vitro data support a role for FOXM1 in ovarian cancer chemotherapy resistance, including platinum-based drugs, taxanes, and PARPi, all of which are currently used to treat ovarian cancer. Conversely, FOXM1 expression was recently reported to be a predictor of increased efficacy for Chk1 and WEE1 inhibitors, which are in clinical testing for ovarian cancer [367,368,369]. It is thus highly relevant to assess FOXM1 as a biomarker for responsiveness to chemotherapeutic agents in current use as well as in clinical trials for ovarian cancer patients.

Transcription factors have traditionally been considered difficult to therapeutically target. However, new strategies have recently been introduced, including disrupting essential protein–protein interactions and promoting targeted proteasomal degradation [370]. Several FOXM1i have been reported, and many appear to bind to and destabilize FOXM1, although their specificity to FOXM1 versus other potential cellular targets requires further characterization. Novel approaches, such as proteolysis targeting chimaeras (PROTACs), could substantially aid in the effort to degrade oncogenic transcription factors such as FOXM1 with increased specificity [370]. Additional FOXM1-targeting methods that may emerge in the future include siRNA, shRNA, and CRISPR-based approaches [371,372,373], which could potentially be delivered intraperitoneally to gain direct access to ovarian cancer cells. Successful targeting of estrogen receptors (ER) in breast cancer and androgen receptors (AR) in prostate cancer has demonstrated the efficacy of targeting oncogenic transcription factors for cancer therapy.

Moving forward, emphasis should be placed on moving FOXM1i studies out of in vitro settings and into state-of-the-art in vivo ovarian cancer models, with the ultimate goal of initiating phase I clinical trials. In summary, the FOXM1 ovarian cancer field is poised to move into a new era that is focused on determining the in vivo roles of FOXM1 in ovarian cancer biology and conducting the initial clinical assessments of its potential as a therapeutic target in patients.

## Figures and Tables

**Figure 1 cancers-13-03065-f001:**
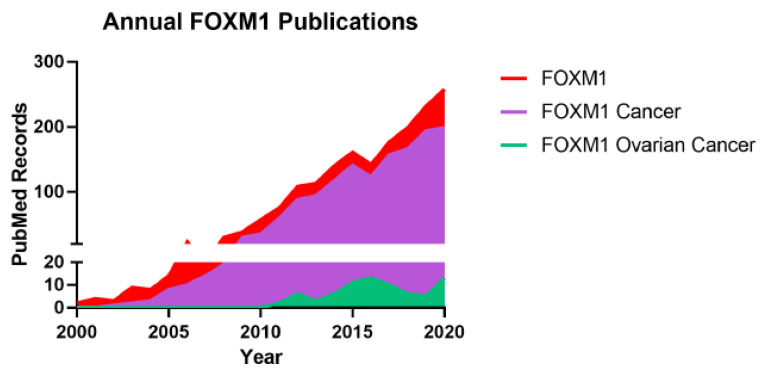
Annual FOXM1 Publications. The data shown were generated from PubMed searches conducted on 04/13/21. The search terms used (all fields) were “FOXM1” (results in red), “FOXM1 cancer” (results in purple), and “FOXM1 ovarian cancer” (results in green). The first “FOXM1 ovarian cancer” PubMed record is the TCGA HGSC study published in 2011 [30].

**Figure 2 cancers-13-03065-f002:**
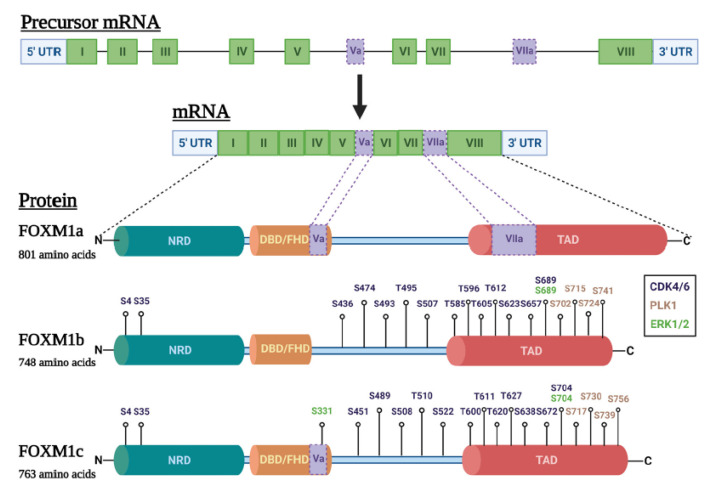
FOXM1 isoforms and phosphorylation sites in ovarian cancer. Top: *FOXM1* precursor mRNA (with introns and exons indicated) followed by *FOXM1* mRNA structure (exons only). Exons shared by all *FOXM1* isoforms are shown in green while alternative exons are shown in light purple. Bottom: Diagram of protein structure of the three major FOXM1 isoforms: (1) FOXM1a, which contains alternative exons Va and VIIa; (2) FOXM1b, which contains no alternative exons; and (3) FOXM1c, which contains alternative exon Va. The three major protein domains are indicated: N-terminal repressor domain (NRD, teal); DNA binding/forkhead domain (DBD/FHD, orange); and transactivation domain (TAD, red). The protein regions corresponding to the alternative exons Va and VIIa are shown in light purple. FOXM1 residues reported to be phosphorylated by three kinases important in ovarian cancer, CDK4/6 (dark blue), PLK1 (tan), and ERK1/2 (green), are indicated. Figure created with BioRender.com.

**Figure 3 cancers-13-03065-f003:**
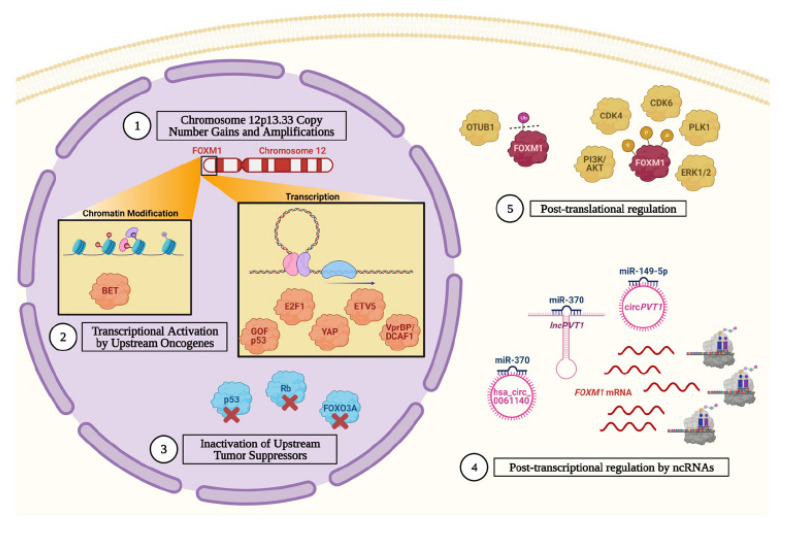
Mechanisms leading to FOXM1 overexpression and activation in ovarian cancer. Several mechanisms that are known to result in FOXM1 overexpression and/or activation in ovarian cancer are shown: (1) Chromosome 12p13.33, where the FOXM1 gene resides, experiences copy number gains and amplifications. (2) Upstream oncogenes (BET, GOF p53, E2F1, YAP, VprBPDCAF1, and ETV5) are overexpressed and upregulate FOXM1 transcription through chromatin modification and promoter activation. (3) Upstream tumor suppressors that regulate FOXM1 transcription (p53, Rb, and FOXO3A) experience inactivation through gene mutation or loss. (4) Post-transcriptional regulation by ncRNAs allow ribosomes to translate high levels of FOXM1 mRNA miR-370 and miR-149-5p, which typically destabilize FOXM1 mRNA, are sponged by lnc*PVT1*, circ*PVT1*, and hsa_circ_0061140. (5) Post-translation regulation stabilizes and activates FOXM1 protein. OTUB1 deubiquitinates FOXM1 and increases protein stability. PI3K/AKT, CDK4, CDK6, PLK1, and ERK1/2 phosphorylate FOXM1, which activates its transcription factor function. Figure created with BioRender.com.

**Figure 4 cancers-13-03065-f004:**
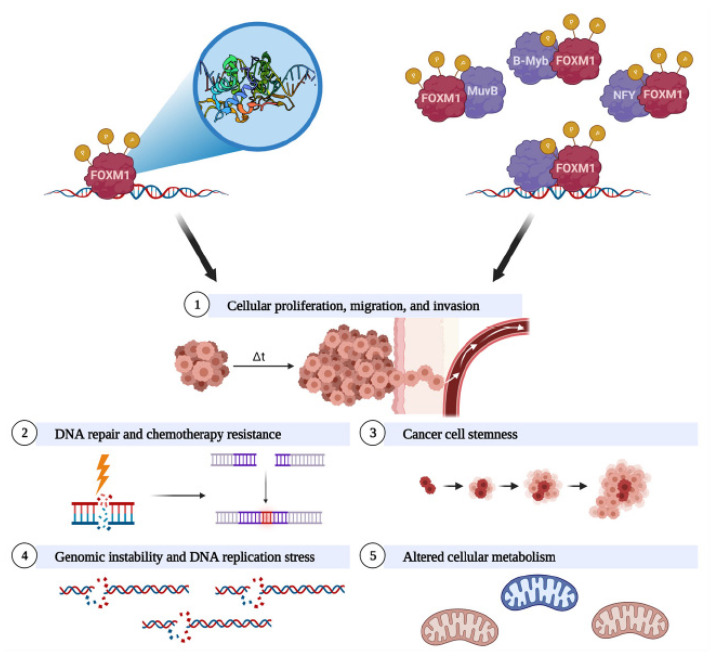
The oncogenic functions of FOXM1 in ovarian cancer. FOXM1 transactivates genes by binding to gene enhancers and promoters, either directly through its DNA binding domain (DBD) or indirectly by interacting with other transcription factors (B-Myb, MuvB, and NFY). Through these two mechanisms, FOXM1 has been shown to promote several oncogenic phenotypes in ovarian cancer, including: (1) cellular proliferation, migration, and invasion; (2) DNA repair and chemotherapy resistance; (3) cancer cell stemness; (4) genomic instability and DNA replication stress; (5) altered cellular metabolism. Figure created with BioRender.com.

**Table 1 cancers-13-03065-t001:** FOXM1 isoforms and variants.

Isoform Name	Structure	Known Function	Size	RefSeq ID	Ensembl ID	UniProt ID	References
**Well-Characterized FOXM1 Isoforms**							
FOXM1a	Includes alternative exons Va and VIIa	Not transcriptionally active	801 aa	NM_202002	ENST00000342628	Q08050-3	[13,14,28,29]
FOXM1b	Omits alternative exons Va and VIIa	Transcriptionally active	748 aa	NM_202003	ENST00000361953	Q08050-2	[13,14,28,29]
FOXM1c	Includes alternative exon Va; omits alternative exon VIIa	Transcriptionally active	763 aa	NM_021953	ENST00000359843	Q08050-1	[13,14,28,29]
**Other Reported FOXM1 Isoforms**							
FOXM1b1	Omits alternative exons Va and VIIa; omits alanine residue at the beginning of exon III; includes glutamine residue at the end of exon V	Transcriptionally active with functions similar to FOXM1b	748 aa	NM_001243088	ENST00000627656	A0A0D9SFF0	[33]
FOXM1b2	Omits alternative exons Va and VIIa; omits alanine residue at the beginning of exon III	Transcriptionally active with functions similar to FOXM1b	747 aa	NM_001243089	N/A	N/A	[33]
FOXM1d	Includes alternative exon VIIa; omits alternative exon Va	Not transcriptionally active; binds directly to oncogenic proteins	786 aa	N/A	N/A	A0A2P9DTZ0-1	[31,34,35]
FOXM1 variant gAug10	No evidence at the protein level	N/A	N/A	N/A	ENST00000536066	N/A	[36]
FOXM1 variant lAug10	No evidence at the protein level	N/A	N/A	N/A	N/A	N/A	[36]

**Table 2 cancers-13-03065-t002:** FOXM1 target genes and their functions.

FOXM1 Target (Gene)	Known Oncogenic Mechanism	References
**Cellular Proliferation**		
Cyclin B1 (*CCNB1*)	Cyclin protein that promotes mitosis	[127,222]
S-phase kinase-associated protein 2 (*SKP2*)	F-box protein that mediates cell cycle entry and G1/S transitions	[127]
Polo-like kinase 1 (*PLK1*)	Protein kinase that mediates mitosis and cytokinesis	[127]
Cell division cycle 25B (*CDC25B*)	Tyrosine protein phosphatase that mediates cell cycle progression and mitosis	[222]
Centrosomal protein 55 (*CEP55*)	Mitotic phosphoprotein that mediates cytokinesis	[222]
Centrometere protein F (*CENPF*)	Microtubule-binding protein and mediates cell division	[222]
DNA topoisomerase II Alpha (*TOP2A*)	DNA topoisomerase that mediates DNA transcription and replication and chromosome condensation and segregation	[222]
Cyclin F (*CCNF*)	F-box protein that mediates the stability of proteins that regulate cell cycle and genome stability	[223]
Protein regulator of cytokinesis 1 (*PRC1*)	Microtubule-associated protein essential for cytokinesis (related to mitosis-related genes in ovarian cancer)	[223]
Homeobox DLX-1 (*DLX1*)	Transcription factor that modulates the TGF-β1/SMAD4 signaling pathway in ovarian cancer	[224]
Proliferation cell nuclear antigen clamp-associated factor (*PCLAF*)	PCNA-binding protein that regulates DNA repair, cell cycle progression, and proliferation (and activates the PI3K/AKT/mTOR signaling pathways in ovarian cancer)	[225]
Kinesin-like protein KIF20A (*KIF20A*)	Kinesin protein that participates in cytokinesis and intracellular transportation	[223]
**Cellular Migration and Invasion**		
Cyclin F (*CCNF*)	F-box protein that mediates the stability of proteins that regulate cell cycle and genome stability	[223]
Protein regulator of cytokinesis 1 (*PRC1*)	Microtubule-associated protein essential for cytokinesis (related to mitosis-related genes in ovarian cancer)	[223]
Matrix metalloproteinase 2 (*MMP2*)	Metalloproteinase that mediates extracellular matrix degradation	[222]
Homeobox DLX-1 (*DLX1*)	Transcription factor that modulates the TGF-β1/SMAD4 signaling pathway in ovarian cancer	[224]
Proliferation cell nuclear antigen clamp-associated factor (*PCLAF*)	PCNA-binding protein that regulates DNA repair, cell cycle progression, and proliferation (and activates the PI3K/AKT/mTOR signaling pathways in ovarian cancer)	[225]
Kinesin-like protein KIF20A (*KIF20A*)	Kinesin protein that participates in cytokinesis and intracellular transportation	[223]
Cytokeratin-5 (*KRT5*)	Filament protein that is found in FTE stem cells and serous ovarian cancer (may promote stemness)	[226,227]
Cytokeratin-7 (*KRT7*)	Filament protein that promotes cell–matrix adhesion and extracellular matrix degradation in ovarian cancer	[226,228]
β-catenin (*CTNNB1*)	Transcriptional co-regulator protein and adaptor protein for cell adhesion that contributes to ovarian cancer metastasis, stemness, chemoresistance, angiogenesis, and immune evasion	[229,230]
Integrin β1 (*ITGB1*)	Integrin protein that facilitates the adhesion of ovarian cancer spheroids	[231]
Integrin αV (*ITGAV*)	Integrin protein that facilitates the adhesion of ovarian cancer spheroids	[231]
Integrin α5 (*ITGA5*)	Integrin protein that facilitates the adhesion of ovarian cancer spheroids	[231]
Lamin B1 (*LMNB1*)	Nuclear lamina protein that facilitates the adhesion of ovarian cancer spheroids	[231]
Fibronectin 1 (*FN1*)	Extracellular matrix glycoprotein that facilitates the adhesion of ovarian cancer spheroids	[231]
**Chemotherapy Resistance and DNA Repair**		
Exonuclease 1 (*EXO1*)	Homologous DNA damage repair protein	[232]
Proliferation cell nuclear antigen clamp-associated factor (*PCLAF*)	PCNA-binding protein that regulates DNA repair, cell cycle progression, and proliferation (and activates the PI3K/AKT/mTOR signaling pathways in ovarian cancer)	[225]
Protein regulator of cytokinesis 1 (*PRC1*)	Microtubule-associated protein essential for cytokinesis (related to mitosis-related genes in ovarian cancer)	[223]
Cyclin B1 (*CCNB1*)	Cyclin protein that promotes mitosis	[233]
BRCA1 (*BRCA1*)	Homologous DNA damage repair protein	[233,234]
BRCA2 (*BRCA2)*	Homologous DNA damage repair protein	[234]
RAD51 (*RAD51*)	Homologous DNA damage repair protein	[233,234]
Fanconi anemia group F protein (*FANCF*)	Homologous DNA damage repair protein	[233]
RAD51 paralog D (*RAD51D*)	Homologous DNA damage repair protein	[233]
Fanconi anemia group D2 protein (*FANCD2*)	Homologous DNA damage repair protein	[233]
**Altered Cellular Metabolism**		
Glucose transporter 1 (*GLUT1*)	Glucose transport protein that promotes aerobic glycolysis in ovarian cancer	[235]
Hexokinase 2 (*HK2*)	Glycolytic enzyme that promotes aerobic glycolysis in ovarian cancer	[235]

**Table 3 cancers-13-03065-t003:** FOXM1 inhibitors and their effects on ovarian cancer phenotypes.

Effect on Ovarian Cancer Cell Phenotype	Concentration	Assays	References
**Thiostrepton**			
Reduced cellular proliferation/viability	0.1–20 µM	XTT, AlamarBlue, sulforhodamine B, MTT	[136,205,229,233,260]
Reduced cellular proliferation/viability of patient ascites cells ex vivo when used alone and in combination with paclitaxel and cisplatin	1–20 µM	Sulfohodamine B	[260]
Reduced cellular proliferation/viability synergistically when used in combination with 1 µM cisplatin	2.5–10 µM	AlamarBlue	[136]
Reduced cellular proliferation/viability by sensitizing cisplatin-resistant cells to cisplatin	0.5–1 µM	MTT	[240]
Reduced cellular proliferation/viability by sensitizing rucaparib-resistant cells to rucaparib	0.1–1.25 µM	Sulfohodamine B	[233]
Reduced cellular migration	5–20 µM	Transwell	[154,205,206,229]
Reduced cellular invasion	5–20 µM	Matrigel transwell	[154,205,206,229]
Reduced colony formation	5–10 µM	Clonogenic	[229]
Reduced colony formation synergistically when used in combination with 2.5 µM FH535 (β-catenin inhibitor)	5 µM	Clonogenic	[229]
Reduced colony formation by sensitizing PARPi-resistant cells to PARPi	0.5–1 µM	Clonogenic	[233]
Slowed wound closure rate	5–10 µM	Wound healing	[206]
Induced apoptosis	1–10 µM	qRT-PCR, western blot, annexin-V/propidium iodide flow cytometry, caspase-3 activity	[136,229,233,260]
Induced apoptosis synergistically when used in combination with 2.5 µM FH535 (β-catenin inhibitor)	5 µM	Annexin-V/propidium iodide flow cytometry	[229]
Induced DNA damage	7.5–10 µM	Alkaline comet	[233]
Induced PARP1 trapping onto chromatin when combined with Olaparib	5–10 µM	PARP trapping	[233]
Reduced sphere formation	1 µM	Spheroid formation	[295]
Decreased HUVEC tube formation and VEFG secretion	5–10 µM	HUVEC tube formation, ELISA	[229]
Decreased *MMP-9* and *PLAUR* gene expression levels	5–10 µM	Sem-quantitative RT-PCR	[206]
Decreased NOTCH1 protein expression levels	1 µM	Western blot	[295]
Decreased active β-catenin, overall β-catenin, TCF4, cyclin D1, cMYC, uPAR, VEGF, MMP-9, and MMP-2 protein expression levels when used alone and in combination with FH535 (β-catenin inhibitor)	5 µM	Western blot	[229]
Reduced tumor size in mice	200–300 µM/kg, 20–50 mg/kg	Cell line-derived xenograft	[205,229,231,240]
Reduced tumor size in mice when used in combination with cisplatin	50 mg/kg	Cell line-derived xenograft	[240]
Reduced tumor size in mice when used in combination with latanib	20 mg/kg	Cell line-derived xenograft	[231]
Reduced tumor size in mice when used in combination with FH535 (β-catenin inhibitor)	20 mg/kg	Cell line-derived xenograft	[229]
Increased overall survival in mice when used in combination with latanib	20 mg/kg	Cell line-derived xenograft	[231]
Reduced number of tumor spheroids in the peritoneal fluid in mice when used alone and used in combination with latanib	20 mg/kg	Cell line-derived xenograft	[231]
Reduced cellular proliferation and induced apoptosis in patient tumors grown ex vivo alone, in combination with olaparib, and in combination with carboplatin	3 µM	Immunofluorescence on fixated tissue	[234]
**FDI-6**			
Reduced cellular proliferation/viability	1–30 µM	Not specified, cell counting kit-8 and microscopic imaging analysis	[362,365]
Reduced cellular proliferation/viability when used in combination with tipifarnib, sapatinib, or rottlerin	3–10 µM	Cell counting kit-8 and microscopic imaging analysis	[365]
Increased N-Ras protein expression	1–10 µM	Western blot	[365]
Decreased p-PKCδ and HER3 protein expression	1–10 µM	Western blot	[365]
**7-difluoromethoxyl-5,4-di-n-octyl genistein (DFOG)**			
Reduced cellular proliferation/viability	1–10 µM	MTT	[360]
Reduced colony formation	1–10 µM	Clonogenic	[360,366]
Induced G2/M-phase cell cycle arrest	1–10 µM	Cell cycle analysis	[360]
Induced apoptosis	1–10 µM	Histone/DNA ELISA, propidium iodide flow cytometry	[360]
Reduced sphere formation	1–10 µM	Spheroid formation	[366]
Decreased CD133, CD44, ALDH1, and NF-κBp65 protein expression levels	1–10 µM	Western blot	[366]
Decreased phosphorylation of AKT, ERK1/2, and FOXO3A	3–10 µM	Western blot	[366]
***N*-phenylphenanthren-9-amine**			
Reduced cellular proliferation/viability	0.01–10 µM	Sulforhodamine B	[321]

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
