# Peer review of "FOXM1: A Multifunctional Oncoprotein and Emerging Therapeutic Target in Ovarian Cancer"

_cancers, 2021, doi:10.3390/cancers13123065_

Round 1
Reviewer 1 Report
The review addresses a very important topic which is much needed in the field of Ovarian cancer. However, in its present form, there are several aspects which require attention. If these are rectified, I support the article for publication.
-no discussion of tumor microenvironment (TME) which is a MAJOR part of OvCa. please include a section in the paper: contribution of FOXM1 to TME including TAM-OvCa, CAF-OvCa, adipocyte-OvCa interactions (does not need to be huge section, but definitely must be included!)
-section of cellular metabolism needs to be expanded significantly. writers should have discussion of effect on lipid metabolism, glycolysis and mito respiration (perhaps a paragraph on each). it is recommended to create a nice diagram of the effects of FOXM1 on metabolic pathways as a replacement of figure 4 (see below for figure 4 recommendations)
-article overall requires more citations and citations that are recent to strengthen content and add to novelty of paper.
-table 1 should include some characteristic or function of each isoform, please include after size column
-remove the first figure. there is no need for it
-figure 4 is more of a graphical abstract not a figure in itself. it is inappropriately placed in the middle of the paper. move it to the top as a graphical abstract OR take out
strong work, well on its way... but please revise accordingly and resubmit.
Reviewer 2 Report
The review of Liu et al. gives an extensive overview of FOXM1 in ovarian cancer. The first chapter describes the general FOX protein family, the history of FOXM1 as well as its functional properties. The short second chapter gives insight to challenges in ovarian cancer, while the third chapter focuses on FOXM1 regulation in ovarian cancer and the fourth chapter treats the topic of the oncogenic function of FOXM1. Chapter five is dedicated to the clincal translation, followed by the conclusion highlighting the importance of FOXM1 in the clinical setting, especially for ovarian cancer. The review is well written and summarizes an intensive amount of data. The authors themselves contributed significantly to this field of research and draw a good picture form the discovery of FOXM1 until now.
Major Comments:
- The chapters are well structured and the headings are descriptive. Although a paragraph at the end of each subchapter tries to conclude the main point, it would be helpful to the reader if the main conclusion would be indicated in the headings.
- The extensive literature given in this review is overwhelming. It is suggested, that the authors, rather than giving a summary of many publications somehow related to ovarian cancer and FOXM1, concentrate on the findings clearly shown in ovarian cancer. Several times results of gastric cancer, glioma cell lines are related to – until now- minor results reached in ovarian cancer. For example in chapter 3.3 it seems that the authors discuss findings of other entities to discuss their own research.
- Further it would be welcome, if the authors would discriminate the relevance of their cited literature. It is mentioned that cell culture experiments are not mirroring the complex clinical situation.
- The main conclusion drawn by this review is the potential of FOXM1 and its inhibitors in the clinical setting. The autors reviewed the literature of the main FOXM1 inhibitors and clearly state that more in vivo and ex vivo data are necessary. The concluding remark that phase I clinical trials should follow, however, needs to be handled with caution as FOXM1 inhibition also has impact on systemic implications as for example the maintenance of hematopietic stem cells.
Reviewer 3 Report
- The current review article on FOXM1 by Liu et al., is a very well rendered manuscript giving a comprehensive and complete information on ovarian cancer articulated by FOXM1.
- The authors have provided seamless flow making it technically flawless and leaving no scope for technical criticism.
- However, since FOXM1 belongs to inhibition of apoptosis (IAP), the authors are requested to include a section on this to make it more perfect manuscript.
- All the other aspects are satisfactory.
Round 2
Reviewer 1 Report
Adequate revisions have been made. Strong and impactful work, well done! I support article for publication.
Reviewer 2 Report
It is a great review, many scientists will benefit from.